Resource

# TaoChongBao: a large-scale *C. elegans* missense variant database bridging worm and human genomes

Ming Li[1,2,3,4,5,6,*], Shimin Wang[1,3,4,5,6,*], Yongping Chai[1,3,4,5,6], Zhengyang Guo[1,3,4,5,6], Zi Wang[1,3,4,5,6], Zhe Chen[1,3,4,5,6], Kexin Lei[1,3,4,5,6], Jingyi Ke[1,3,4,5,6], Xingshen Huang[1,3,4,5,6], Guanghan Chen[1,3,4,5,6], Peng Huang[1,3,4,5,6], Kaiming Xu[1,3,4,5,6], Zijie Shen[1,3,4,5,6], Wei Li[7], Guangshuo Ou[1,2,3,4,5,6]

We generated and sequenced 12,069 viable *Caenorhabditis elegans* strains produced by ethyl methanesulfonate mutagenesis, identifying 20,315,536 variants, including 541,102 unique missense mutations across 20,914 genes. Most strains exhibit resistance to the anti-nematode drug ivermectin, whereas some others display phenotypes like dumpy morphology, uncoordinated movement, multivulva formation, and blistered cuticle. To organize and visualize this resource, we developed TaoChongBao, an open-access database and strain repository that integrates *C. elegans* mutation data with AlphaMissense-predicted pathogenicity scores and ClinVar clinical annotations. TaoChongBao enables users to explore worm missense variants, identify conserved residues corresponding to human pathogenic sites, and access viable strains for experimental validation. Compared with the previous Million Mutation Project in *C. elegans*, TaoChongBao expands mutation coverage over 20-fold and emphasizes amino acid–altering variants. This resource provides a scalable platform for functional residuomics, variant interpretation, and comparative analyses between *C. elegans* and human genomes.

## Introduction

Single-nucleotide substitutions that change one amino acid—known as missense mutations—can profoundly influence protein structure, stability, and interaction networks. Such substitutions constitute a major class of genetic variation in both health and disease. Understanding the functional consequences of missense mutations is therefore a fundamental goal of molecular biology and precision medicine (1, 2, 3, 4, 5). Recent computational advances, such as AlphaMissense, have predicted the pathogenic potential of all

71 million possible single–amino acid substitutions in the human proteome. However, translating these predictions into biological understanding requires scalable experimental models that can evaluate the effects of these substitutions in vivo (2). Current approaches, such as CRISPR-based genome editing, enable researchers to manipulate the genome of cell lines and animal models with ease and accuracy, thereby generating specific codon changes and individual missense variants. However, these methods typically produce one mutation at a time, and necessitate several months and substantial costs (6, 7). A complementary strategy is needed to generate, catalog, and experimentally access large numbers of missense mutations efficiently.

Chemical mutagenesis using ethyl methanesulfonate (EMS) provides such a high-throughput route. Since Sydney Brenner's pioneering work in the 1960s, EMS mutagenesis has been a cornerstone of *Caenorhabditis elegans* genetics (8, 9, 10). Each mutagenized worm typically carries hundreds of single-nucleotide changes, including dozens of missense variants (11, 12, 13), and the simplicity of nematode culture allows the generation of vast mutant populations at minimal cost (10, 14). With recent advances in high-throughput whole-genome sequencing, it has become feasible to systematically identify and analyze these variants across thousands of strains (13, 15, 16, 17).

Here, we present a large-scale *C. elegans* EMS mutagenesis and sequencing initiative comprising more than 12,069 strains and yielding 20,315,536 variants—about 20 times the scale of the Million Mutation Project (13). This includes 541,102 unique missense mutations across nearly all coding genes. We developed TaoChongBao (https://oulab.life.tsinghua.edu.cn/homopatho/index.php), a public database integrating this mutation resource with AlphaMissense predictions (2) and ClinVar annotations (4), enabling users to query conserved worm–human missense sites, visualize pathogenicity information, and access strains carrying relevant variants. Beyond its scale, several features distinguish this resource. Despite heavy mutational loads, all strains remain viable, revealing broad tolerance to single–amino acid substitutions in essential genes and

[1]State Key Laboratory of Membrane Biology, Beijing, China [2]State Key Laboratory of Membrane Biology, Membrane Structure and Artificial Intelligence Biology Branch, Hangzhou, China [3]Tsinghua-Peking Center for Life Sciences, Beijing, China [4]Beijing Frontier Research Center for Biological Structure, Beijing, China [5]McGovern Institute for Brain Research, Beijing, China [6]Artificial Evolution Summer School and Winter Camp in the School of Life Sciences, and MOE Key Laboratory for Protein Science, Tsinghua University, Beijing, China [7]State Key Laboratory of Membrane Biology, School of Basic Medical Sciences, Tsinghua University, Beijing, China

Correspondence: guangshuoou@tsinghua.edu.cn
*Ming Li and Shimin Wang contributed equally to this work

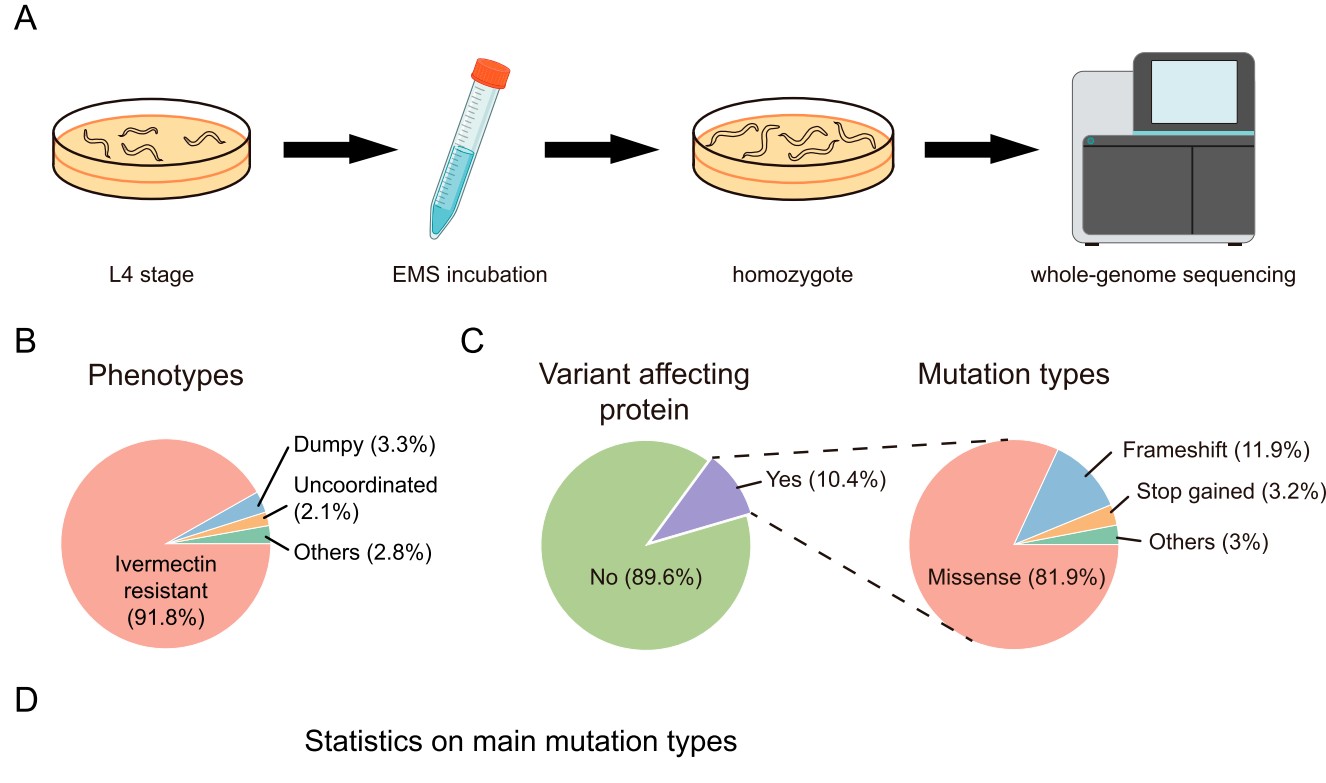

**Figure 1. Generation of *C. elegans* mutant collection.**
**(A)** Flowchart depicts the generation of *C. elegans* mutants. Late L4 animals were mutagenized with EMS. F2 progenies were singled and propagated, and F3 homozygous mutants were selected for whole-genome sequencing. **(B)** Phenotypes observed in the *C. elegans* mutant collection. **(C)** Mutation types identified in the *C. elegans* mutant collection. **(D)** Statistics for the three major classes of mutations.

providing insight into mutational robustness. Our screen also incorporated selection with ivermectin, enriching for mutants with ciliary or ion channel defects relevant to neurobiology and drug response (18, 19, 20, 21). By mapping conserved residues between worm and human proteins, this dataset offers an experimental platform to interpret human variants of uncertain significance. Together, these advances establish *C. elegans* as a scalable and economical model for functional residuomics and variant interpretation.

## Results

### Large-scale EMS mutagenesis and whole-genome sequencing of *C. elegans* strains

Following an established EMS mutagenesis protocol (Fig 1A; see details in the Materials and Methods section), we generated

12,069 viable *C. elegans* strains through successive rounds of mutagenesis, selection, and isolation. To reduce the possibility of spontaneous suppressor mutations arising during strain maintenance, we minimized laboratory passaging. Specifically, each mutant line was cultured for no more than one to two additional generations before sequencing and cryopreservation after phenotype confirmation in the F3 generation. Notably, 78.4% of these strains exhibit a diverse array of phenotypes, including dumpy morphology, uncoordinated movement, multivulva formation, blistered cuticle, and, most commonly, resistance to the anti-nematode drug ivermectin (18, 19, 20, 21) (Fig 1B).

Whole-genome sequencing (WGS) was performed for all these strains, achieving an average coverage depth exceeding 15 per genome. Across all strains, we identified 20,315,536 variants distributed throughout the *C. elegans* genome. Consistent with previously reported EMS mutagenesis rates (11, 12, 13), each strain harbored an average of ~1,700 variants, including ~144 missense mutations that affect protein-coding sequences. Among all

detected variants, 89.6% were located in non-coding regions—including introns, untranslated regions, and intergenic sequences—whereas 10.4% resided within protein-coding exons. Within the coding subset, 81.9% were missense mutations, followed by frameshift (11.9%), nonsense (3.2%), and other categories such as splice site and stop-loss mutations (3%) (Fig 1C). In total, 541,102 unique missense mutations were mapped across 20,914 *C. elegans* genes, corresponding to an average of 26 distinct amino acid substitutions per gene (Fig 1D). Notably, all strains remained viable, implying that most single–amino acid substitutions, even in essential genes, do not compromise viability, thus offering a broad collection of functionally tolerant protein variants.

## Construction of the TaoChongBao database

To effectively organize and leverage this extensive genetic dataset, particularly as a resource for genetic disease research, we developed the TaoChongBao database (Fig 2). First, we retrieved *C. elegans*–human ortholog pairs from the OrthoList 2 project (22), obtaining 28,298 pairs, and supplemented them with an additional 4,572 pairs manually clustered by MMseqs2 (23). We then collected protein sequences for the corresponding *C. elegans* genes from WormBase (24), and human genes from UniProt (25). Recognizing the probability that multiple protein isoforms can be encoded by single gene in both species, we retained all *C. elegans* isoforms presented in our sequencing data, whereas for human genes, we retained only the isoform designated as "canonical" by UniProt. This selection yielded 13,790 *C. elegans* isoforms and 11,137 human isoforms. Subsequently, we aligned each isoform pair using the Clustal W package with the BLOSUM62 scoring matrix (26), revealing nearly 3,690,301 conserved amino acid residues between *C. elegans* and human proteins (Fig 3A).

Next, we integrated predicted pathogenicity data from the AlphaMissense (2), which encompasses about 71 million potential single–amino acid substitutions, together with clinically reported variants from ClinVar (4). AlphaMissense provides an in silico prediction that estimates the probability of a missense variant disrupting protein function based on a deep learning model. It gives probabilistic scores but does not incorporate patient-level clinical evidence. In contrast, ClinVar offers clinically curated interpretations of variant pathogenicity, submitted by laboratories and researchers, and supported by diverse lines of evidence, including segregation data, population frequency, functional assays, and clinical observations.

From both sources, we extracted information on substituted amino acids, predicted or reported pathogenicity, associated diseases, and other relevant annotations. We then defined "homologous pathogenic mutation" as a missense mutation occurring at a conserved residue where the altered amino acid is predicted or reported to be pathogenic or likely pathogenic. Notably, our analysis revealed that 89.7% conserved sites contained at least one such homologous pathogenic mutation (Fig 3A).

Furthermore, we structured our sequencing results into a unified database containing essential information such as strain name, phenotypic data, gene, transcript, mutation type, position,

DNA variant, and resulting protein changes. To integrate these data seamlessly with the pathogenicity annotations described above, we introduced a new field to indicate whether each mutation is classified as a homologous pathogenic mutation. From the 541,102 unique missense mutations across our *C. elegans* mutant collection, we found 120,363 (22.2%) mutations occurring at the conserved site (Fig 3A). Approximately half of them are predicted to be pathogenic by AlphaMissense (Fig 3B), and 5.2% (6,255 mutations) are recorded in ClinVar (Fig 3C). Furthermore, 625 mutations are classified as pathogenic or likely pathogenic based on actual clinical observations, underscoring their potential value for the study of human genetic diseases.

Our database therefore supports two major retrieval functions: (a) users can query whether a given missense mutation in *C. elegans* corresponds to a homologous pathogenic mutation in human and, if so, obtain associated pathogenicity and disease information; and (b) users can identify whether any *C. elegans* strain in our strain resource library carries homologous pathogenic mutations of interest.

## The TaoChongBao website

The TaoChongBao website interface is designed to be intuitive and user-friendly. Users can search for either a *C. elegans* or human gene of interest by entering a gene symbol, UniProt identifier, WormBase ID (for *C. elegans*), or HGNC ID (for human) (Fig 4A). The system will automatically recognize the type of input and queries the appropriate dataset. For example, submitting a *C. elegans* gene like "*unc-104*" returns the results shown in Fig 3B. Basic information about "*unc-104*" is displayed in the title bar, including organism, gene symbol, a brief description, and cross-references to other relevant databases (Fig 4B ①). Because some *C. elegans* genes encode multiple isoforms, a dropdown menu lists all available isoforms, albeit in most cases only one isoform is returned (Fig 4B ②). When multiple isoforms exist, they are sorted by the number of available strains carrying mutations in that isoform, with the first entry selected by default. User may also manually choose another isoform. This isoform-selection step is unnecessary when querying a human gene, as only one canonical isoform is included in the database.

After selecting the desired isoform, each human ortholog is displayed as expandable tabs in the main content area (Fig 4B ③). Each tab features a dot plot that visualizes the pathogenicity of all conserved sites between the orthologous pairs, for instance, *unc-104* and KIF1A, as shown in Fig 4B ④. The plot includes a draggable horizontal axis indicating the conserved positions in *C. elegans* and human protein sequences, and a vertical axis indicating the amino acid at each site (* for stop codon). Wild-type residues are denoted by diamond markers, whereas mutated residues appear as round markers. The color of each round marker reflects the AlphaMissense predictions: red for pathogenic, green for benign, and gray for uncertain. Mutations recorded in ClinVar are indicated by a smaller dot centered on the round marker, following the similar color scheme. If a variant is present in our strain resource library, the round marker will be outlined by a yellow square. Hovering over

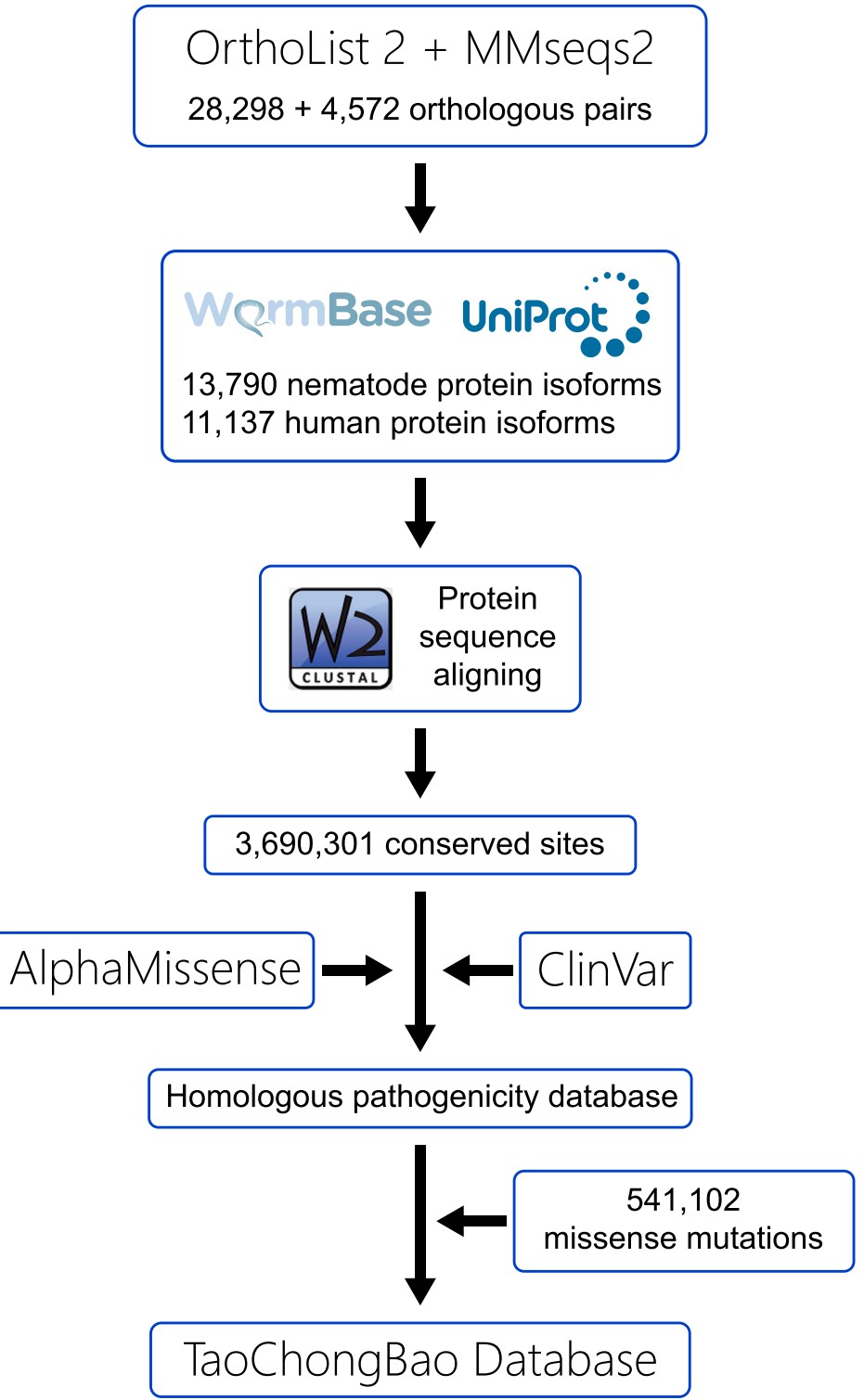

**Figure 2. Workflow for constructing the TaoChongBao database.**
*C. elegans*–human orthologous pairs were retrieved from OrthoList 2 project and clustered by MMseqs2. Corresponding protein isoform sequences were obtained from WormBase and UniProt, then aligned using the Clustal W package, yielding 3,690,301 conserved residues. Pathogenicity data from AlphaMissense and ClinVar were manually curated and integrated with the 541,102 missense mutations identified in our *C. elegans* mutant collection.

any marker displays location and amino acid information of the corresponding variant (Fig 4B ⑤), and clicking on it opens a panel with full annotations from AlphaMissense, ClinVar, and the strain stock (Fig 4C). If one or more strains are available in our resource library, a jump button next to the strain name allows the user to access to a page summarizing all mutations carried by that strain (Fig 4D). In addition, each ortholog tab also includes basic information similar to that in the title bar, along with a simple statistics summarizing the available strains harboring mutations at these conserved sites (Fig 4B ⑥).

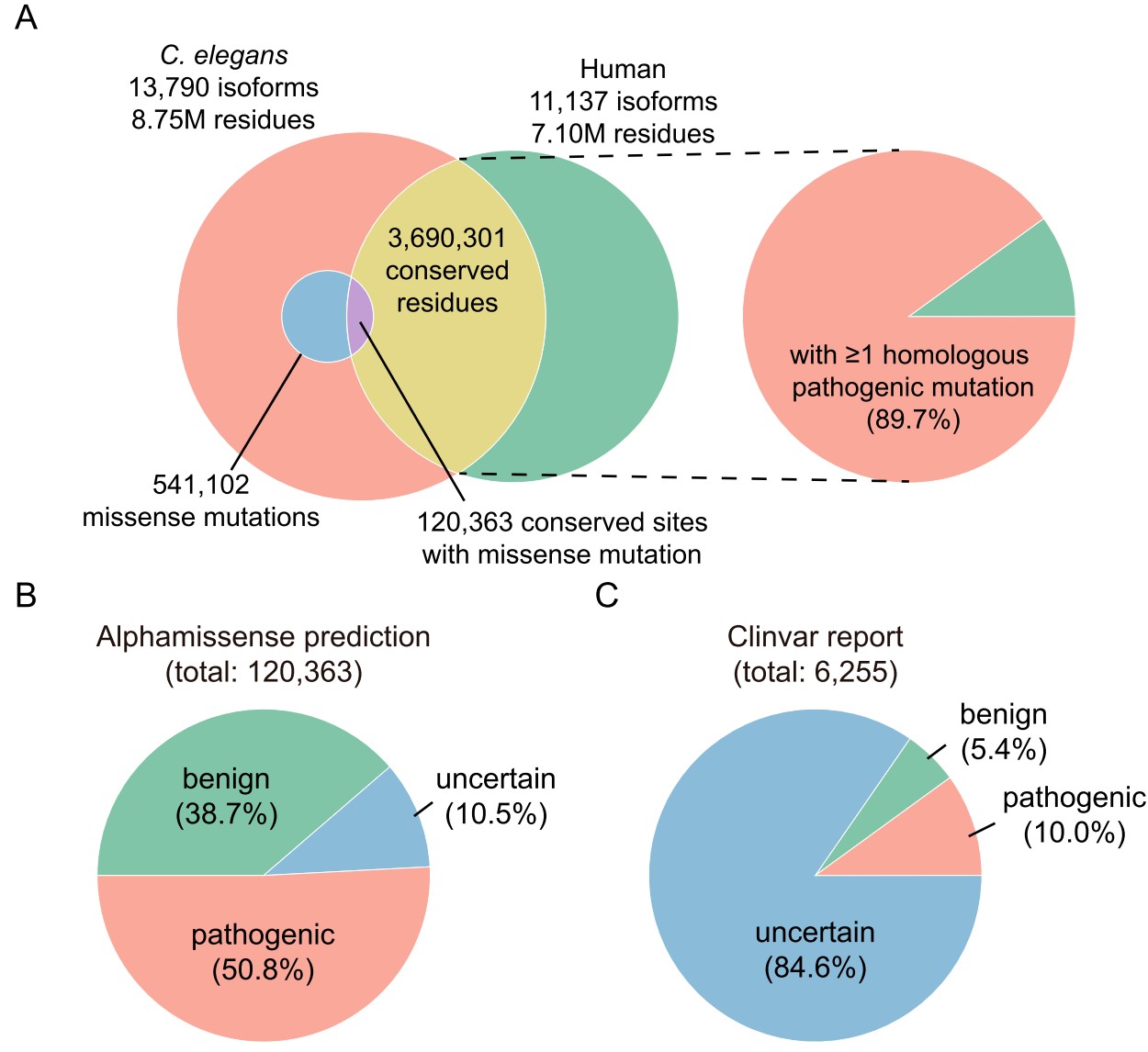

**Figure 3. Integration of pathogenicity data and missense mutations in the *C. elegans* mutant collection.**
**(A)** Venn diagram illustrates the distribution of *C. elegans*–human conserved residues and the 541,102 missense mutations across the *C. elegans* mutant collection. **(B)** Predicted pathogenicity of the missense mutations occurring at conserved sites. **(C)** Reported pathogenicity of the missense mutations occurring at conserved sites.

## Discussion

We report the largest EMS mutagenesis and whole-genome sequencing effort in *C. elegans* to date, generating a comprehensive resource of missense variants across more than 12,069 strains. This dataset includes over 20 million genomic variants, including 541,102 protein-altering missense mutations distributed across 20,914 genes—representing a 20-fold expansion over the previous Million Mutation Project ([13]). To enable broad and effective use of these data, we developed the TaoChongBao database, an integrated platform that supports visualization and querying of mutations, assessment of conservation with human pathogenic residues, and access to viable strains for functional assays. By combining large-scale experimental mutagenesis with both

predictive and clinical annotated pathogenicity information, this resource bridges in silico pathogenicity predictions with in vivo functional validation.

EMS mutagenesis remains one of the most cost-effective and scalable methods for generating genome-wide variant diversity. In our collection, we identified more than 120,363 missense mutations in *C. elegans* that are homologous to human mutations, including 625 that correspond to clinically annotated pathogenic alleles. For comparison, generating even a single *C. elegans* strain via genome-editing approaches (e.g., CRISPR-Cas9) typically requires roughly 1 mo and a minimum cost of US$1,000. Modeling all 625 pathogenic missense variants in our benchmark set would cost US$625,000 using targeted genome editing, and modeling the full set of 120,363 homologous human variants would exceed

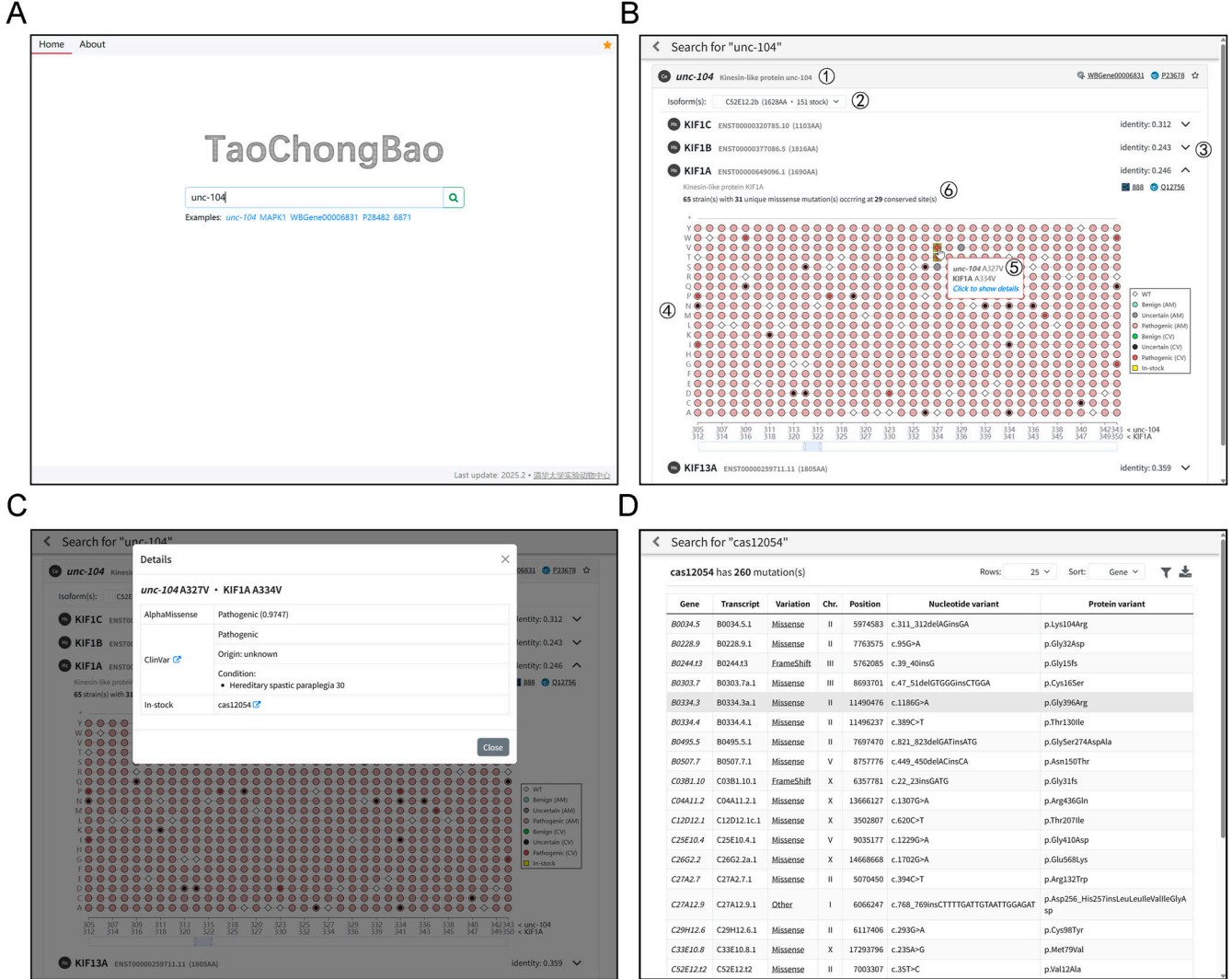

**Figure 4.   Interfaces of the TaoChongBao website.**
**(A)** Home page, allowing user to search for *C. elegans* or human genes with various identifiers. **(B)** Search results page displaying basic information for queried gene and the homologous pathogenicity information between the orthologous pairs. WT, wild type; AM, AlphaMissense; CV, ClinVar. **(C)** Detail panel showing the full annotations from AlphaMissense and ClinVar, and the strain stock. **(D)** *C. elegans* strain page summarizing all mutations carried by the selected strain.

US$120 million—an impractical undertaking. Although EMS mutagenesis is random and cannot guarantee any specific substitution, scaling to thousands of sequenced strains substantially increases the likelihood of capturing many clinically relevant residues. Notably, our entire 12,069 strain collection was generated and sequenced for ~US$240,000, with each strain costing about US$20 (27). These comparisons underscore that large-scale chemical mutagenesis, while complementary rather than alternative to precise editing, offers an exceptionally cost-effective strategy for generating broad allelic series that would be prohibitively expensive to obtain through genome engineering alone, thereby providing a practical platform for functional interpretation of human variation.

Although our strain library provides a large mutant resource, it is important to point out its scope and potential biases. Approximately 70% of the strains were isolated through ivermectin resistance screening, whereas the others were obtained from morphology variants (e.g., dumpy), locomotor defects (e.g., uncoordinated movement), etc., and a smaller set from suppressor screens. As a result, genes involved in ivermectin response and related pathways are expected to be enriched relative to an unbiased mutational spectrum. This enrichment reflects the screening strategies rather than spontaneous selection within the collection. Nevertheless, because EMS induces numerous random mutations, background variants segregate independently during the F2 generation therefore may also become homozygous. Consequently, aside the ivermectin resistance–associated genes, a small subset of background mutations can still become fixed in individual strains. Future expansion of the library using additional selection strategies and non-selective propagation will further broaden the functional diversity of the mutant collection.

Chemical mutagenesis primarily generates G/C-to-A/T conversions, limiting codon coverage (9, 11, 12, 13). Analyses of the Million Mutation Project suggest that EMS mutagenesis is influenced by nucleotide context and chromatin structure, meaning complete residue saturation has not yet been achieved (12, 13). Some mutations, particularly lethal alleles, remain undetected, which can be addressed through temperature-sensitive or conditional genetic screens (28, 29, 30). Nonetheless, this scalable strategy provides a framework for systematically generating and functionally characterizing missense variants, with potential applications in other model organisms and human cell lines amenable to forward genetic screens.

Our strain resource library serves as a versatile tool for diverse applications. It provides genetically diverse strains that allow researchers to explore the relationships between protein sequence and function, including studies on protein folding, stability, and interactions, which are fundamental to understanding molecular biology. It is also meaningful to examine whether mutants in the library exhibit additional phenotypes, such as temperature sensitivity, chemotaxis variant, or chemical hypersensitivity. This may provide an opportunity to identify new genes and mechanisms contributing to these phenotypes. In addition, the resource supports suppressor and enhancer screens, allowing identification of genetic interactions that mitigate the effects of deleterious mutations.

Beyond serving as a large-scale library of sequenced mutants, our platform provides a systematic framework facilitating disease modeling. By integrating mutagenized *C. elegans* variants with cross-species conservation analysis and human pathogenicity annotations, the library enables the identification of homologous pathogenic mutations affecting conserved residues. A representative example is the *unc-104 (R9Q)* variant (31), which was initially identified within our developing mutant collection and prioritized through homologous pathogenicity analysis linking it to the human disease–associated KIF1A (R11Q) mutation. The introduction of this mutation into clean background worms using CRISPR-Cas9 successfully recapitulated morphology and movement phenotypes. This case illustrates how the integration of large-scale mutagenesis, pathogenicity annotation, and targeted genome editing can transform a mutant resource into a predictive platform for modeling conserved, disease-associated missense variants. We expect that continued application of this platform will facilitate dissection of human pathogenic mutations and aid in identifying therapeutic targets.

At present, we have cryopreserved 12,069 *C. elegans* strains, and are willing to share these resources upon request. However, to maintain such a large collection requires dedicated infrastructure, including two ultra-low-temperature freezers, and a technician responsible for inventory management, routine freezing, and recovery. Despite these efforts, the scale of the collection makes it impractical to routinely retest the viability of all frozen strains. Our experience in maintaining this library has highlighted the logistical and financial challenges associated with a centralized repository. Meanwhile, like existing centralized repositories, such as the *C. elegans* Genetics Center or commercial platforms like Addgene, we also face inherent capacity limits and cannot expand indefinitely as the number of unique strains continues to grow. These practical limitations underscore the need for a more scalable and distributed framework: a decentralized, marketplace-style model—analogous to Amazon—where individual laboratories contribute their own sequenced strains, metadata, and annotations, and the broader community can browse, query, and obtain variants of interest. TaoChongBao aims to become such a decentralized platform. We are actively advancing its development and gradually adding features that facilitate resource sharing, enabling laboratories to share resources directly while reducing redundancy, distributing storage burdens, and promoting a sustainable, community-driven system for genetic research.

In conclusion, TaoChongBao provides a comprehensive, community-accessible resource of *C. elegans* missense variants, integrating whole-genome sequencing data with predictive and clinical annotations. The database allows users to visualize mutations, assess conservation with human pathogenic residues, and access viable strains for functional assays. By expanding mutation coverage over 20-fold relative to previous resources, TaoChongBao enables systematic studies of protein function, genetic interactions, and human disease modeling. We anticipate that TaoChongBao will become a widely used platform for functional residuomics, variant interpretation, and experimental genetics in both basic and translational research.

# Materials and Methods

### *C. elegans* strain culture

*C. elegans* strains were maintained as described previously (10), on nematode growth medium (NGM) plates with OP50 feeder bacteria at 20°C. All animal experiments were performed following governmental and institutional guidelines.

### EMS mutagenesis

EMS mutagenesis was performed as described before with some modifications (32, 33). Worms at the late L4 stage were cultured on regular NGM plates seeded with OP50 feeder until the population was just starved. Then, worms from 10 to 20 plates were collected with M9 buffer (3 g $KH_2PO_4$, 6 g $Na_2HPO_4$, 5 g NaCl, 1 ml 1 M $MgSO_4$, $H_2O$ to 1 liter), and incubated with 50 mM EMS buffer at room temperature with continuous rotation for 4 h. After incubation, worms were washed with M9 buffer and dispersed to 200–400 OP50-seeded NGM plates. After 20 h, adult worms were subjected to a bleaching protocol to isolate their eggs (F1 generation). These eggs were then transferred to individual NGM plates at a density of about 10 eggs/plate and cultured at 20°C. When the F2 progeny reached the young adult stage, a single mutant was isolated under a stereomicroscope and transferred to a new NGM plate for screening, for example, ivermectin screening or dumpy/uncoordinated screening. Worm strains that stably transmitted the phenotype to the F3 generation were expanded by one or two generations, and subjected to WGS using an Illumina next-generation sequencer.

## Whole-genome sequencing

Approximately 5,000-20,000 worms were used for genome DNA extraction per sample. Worms were washed off plates with M9 buffer and washed three to four times. After the final wash, the HE worms were resuspended in lysis buffer (100 mM Tris-HCl, 50 mM NaCl, 10 mM EDTA, and 1% SDS, pH 8.5) supplemented with Proteinase K (100 μg/ml) and incubated at 65°C for 2–3 h. RNA was removed by RNase A (50 μg/ml) treatment, followed by phenol–chloroform–isoamyl alcohol extraction. DNA was precipitated with cold ethanol and pelleted by centrifugation at 12,000$g$ for 15 min at 4°C, then washed with 70% ethanol and resuspended in 50–100 μl TE buffer. DNA quantity and quality were assessed using NanoDrop spectrophotometer, Qubit 4 Fluorometer, and agarose gel electrophoresis. To prepare sequencing libraries, a total amount of 300–500 ng DNA per sample was fragmented to ~350 bp using sonication (Covaris system), and then endpolished, A-tailed, and ligated to indexed adapters following the protocols of the NEBNext Ultra DNA Library Prep Kit for Illumina. Libraries were purified using AMPure XP Beads (Beckman Coulter) and enriched by PCR amplification. The size distribution and concentration of libraries were assessed using Agilent 2100 Bioanalyzer and Qubit 4 Fluorometer.

Libraries were pooled and sequenced on an Illumina NovaSeq 6000 Sequencing System to generate ~150-bp paired-end reads, achieving an average genome coverage of at least 20× per sample.

Raw WGS reads (FASTQ file) were quality-controlled using FastQC (v0.12.1), trimmed for adapters and low-quality bases by Trim_galore (v0.6.8), and aligned to WBcel235 of the *C. elegans* genome (https://www.ensembl.org/Caenorhabditis_elegans/Info/Index) by BWA-MEM2 (v2.2.1). All datasets achieved >15× coverage.

After removing putative PCR duplicates using Picard (v2.27.3), variants, including single-nucleotide variants (SNVs), insertion/deletion variants (indels), and multi-nucleotide polymorphisms (MNPs), were called using freebayes (v1.3.6), and annotated with SnpEff (v5.0d). To obtain homozygous SNVs, variants were filtered using vcflib tools (v1.0.3) with the criteria "QUAL > 20 and DP > 5 and AF > 0.8," retaining locations with at least 5× coverage, ≥80% consensus variant allele, and a minimum mapping quality of 20. Essential information for constructing the TaoChongBao database was extracted using a homemade Python script.

## Data Availability

All TaoChongBao resources are accessible through the web portal (https://oulab.life.tsinghua.edu.cn/homopatho/index.php). The data underlying this work can be found at Zenodo (https://zenodo.org/records/18948671).

## Acknowledgements

This work was supported by the following funding programs: the National Key R&D Program of China Grants 2024YFA1307301, 2022YFA1302700, and 2019YFA0508401; the National Natural Science Foundation of China Grants 92254306, 31991190, 32270773, 32470730, 32070706, 32270721, 32430026, 32570812, and 32400610; and the Postdoctoral Fellowship Program of China Postdoctoral Science Foundation GZC20240861. The genetic screens were performed at the 2022–2025 Artificial Evolution Summer School supported by the Tsinghua University and Qingshan Lake Science and Technology City in Hangzhou, China.

## Author Contributions

M Li: conceptualization, data curation, software, formal analysis, funding acquisition, investigation, and writing—original draft, review, and editing.
S Wang: data curation, formal analysis, and investigation.
Y Chai: data curation, formal analysis, funding acquisition, and investigation.
Z Guo: data curation, software, and formal analysis.
Z Wang: data curation, formal analysis, funding acquisition, and investigation.
Z Chen: data curation, formal analysis, and investigation.
K Lei: data curation, formal analysis, and investigation.
J Ke: data curation, formal analysis, and investigation.
X Huang: data curation, formal analysis, and investigation.
G Chen: data curation, formal analysis, and investigation.
P Huang: data curation, formal analysis, and investigation.
K Xu: data curation, formal analysis, and investigation.
Z Shen: data curation, formal analysis, and investigation.
W Li: resources, supervision, funding acquisition, and project administration.
G Ou: conceptualization, resources, supervision, funding acquisition, project administration, and writing—original draft, review, and editing.

## Conflict of Interest Statement

The authors declare that they have no conflict of interest.

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
