## [Reviewer comments · Life Science Alliance]

TaoChongBao: A Large-Scale *C. elegans* Missense Variant Database Bridging Worm and Human Genomes

Ming Li, Shimin Wang, Yongping Chai, Zhengyang Guo, Zi Wang, Zhe Chen, Kexin Lei, Jingyi Ke, Xingshen Huang, Guanghan Chen, Peng Huang, Kaiming Xu, Zijie Shen, Wei Li, and Guangshuo Ou

DOI: <https://doi.org/10.26508/lsa.202603631>

Corresponding author(s): Guangshuo Ou, Tsinghua University

Review Timeline:

Submission Date:	2026-01-14
Editorial Decision:	2026-02-12
Revision Received:	2026-03-04
Accepted:	2026-03-12

Scientific Editor: Tim Fessenden

Transaction Report:

February 12, 2026

RE: Life Science Alliance Manuscript #LSA-2026-03631

Dr. Ming Li
Tsinghua University
College of Life Sciences
D217 Zhengyutong Medical Building
Tsinghua University
Beijing 100084
China

Dear Dr. Li,

Thank you for submitting your manuscript entitled "TaoChongBao: A Large-Scale *C. elegans* Missense Variant Database Bridging Worm and Human Genomes" to Life Science Alliance. This manuscript describing a web resource to access and analyze results from the largest ever bank of mutant *C. elegans* lines was reviewed by three expert referees.

As you will see in their reports appended to this letter, all three reviewers voiced clear enthusiasm for this work. Each reviewer made suggestions to improve the text and the associated web platform, and here we draw your attention to the question of ivermectin resistance noted by Reviewers 1 and 2. We invite you to address these and all other suggested improvements in the manner of your choice. Given their unanimous support for publication without major reservations or concerns, we would be happy to publish your paper in Life Science Alliance pending these text changes and final revisions necessary to meet our formatting guidelines.

MANUSCRIPT ORGANIZATION AND FORMATTING:

To avoid unnecessary delays in the acceptance and publication of your paper, please read the following information carefully. Full guidelines are available on our Instructions for Authors page, <https://www.life-science-alliance.org/authors>

- Please upload your main manuscript text as an editable doc file.
- Please upload your figures as single files.
- Please add the X and Bluesky handles of your host institute/organization, as well as your own, and/or one of the authors, in our system.
- The corresponding author must match between the system and the manuscript file. Please correct accordingly.
- Please add an ORCID ID for the corresponding author -- they should have received instructions on how to do so.
- Please add an Author Contributions section to your main manuscript text. The contributions selected for Wei Li do not qualify them for authorship. Please either update the contributions in our system and in the Author Contributions section of the manuscript, or let us know if this author should be removed (and added to the acknowledgment section).
- It is recommended to exclude figures from the manuscript text and upload them separately.
- Please add a Conflict of Interest statement to your main manuscript text.
- Please add your figure legends to the main manuscript text after the references section and remove them from the figures.

LSA encourages authors to provide a 30-60 second video where the study is briefly explained. We will use these videos on social media to promote the published paper and the presenting author (for examples, see <https://docs.google.com/document/d/1-UWCfbE4pGcDdcgzcmiuJl2XMBJnxKYeqRvLLrLSo8s/edit?usp=sharing>). Corresponding or first-authors are welcome to submit the video. Please submit only one video per manuscript. The video can be emailed to

contact@life-science-alliance.org

FINAL FILES:

The following items are required for acceptance.

The license to publish form must be signed before your manuscript can be sent to production. A link to the license to publish form will be available to the corresponding author only. Please take a moment to check your funder requirements.

Thank you for your attention to these final processing requirements. Please revise and format the manuscript and upload materials as soon as you are able.

Thank you for this interesting contribution to the literature. We look forward to publishing your paper in Life Science Alliance.

Sincerely,

Reviewer #1 (Comments to the Authors (Required)):

Li, Wang, et al describe a massive collection of viable *C. elegans* mutant strains in which they have used whole genome sequencing to map millions of mutations throughout their genomes. Notably, they have classified all of these mutations based on their presence in coding or non-coding regions, and they have identified over 500,000 mis-sense mutations across nearly all *C. elegans* genes (with an average of 26 mis-sense mutations per affected gene, a remarkable accomplishment). Furthermore, they have established a user-friendly database that integrates the mutation collection with Alpha-fold-based pathogenicity scores and clinical annotations. They also identify a subset of the mutant strains that are ivermectin-resistant and thus likely defective in ciliary or ion channel functions. This is an extremely impressive viable and well described mutant collection and database that are generally available and should provide numerous opportunities for other researchers to explore human disease gene variant function using *C. elegans* as a model system. This manuscript without question warrants publication as written, but the following minor comments and suggestions are offered for the authors' consideration.

1. Growing viable but somewhat sickly mutant strains can inadvertently lead to the appearance of spontaneous suppressor mutations that improve viability over multiple generations of growth. It might be worth noting roughly how many generations the mutant strains were grown before being frozen, assuming they were all frozen.
2. The identification of ivermectin-resistant strains is very nice, and the authors might want to include mention of this in their abstract to make these mutant strains more prominent and visible to interested researchers.
3. I might have missed it, but I did not see anywhere in the manuscript text indicating that the authors have indeed frozen all of these strains, and if the strains are available upon request. If so, this should be mentioned in the paragraph that points out the limitations of the stock center when it comes to handling such a large mutant collection.

4. While beyond the scope of this manuscript, it would be interesting for the authors to grow all of their strains at 25°C to see if any might be temperature-sensitive, with enough function at 20°C to make the strains homozygous viable at that lower temperature. Also, it would be very interesting if the authors could identify a few mis-sense mutations that are highly likely to be relevant to human disease and have visible phenotypes, and then see if using CRISPR to create the mutation in an otherwise wild-type background leads to the same visible phenotype. Alternatively, the authors might mention this possible strategy for future research in the Discussion section.

Reviewer #2 (Comments to the Authors (Required)):

In this manuscript Li and colleagues used EMS mutagenesis to generate a large number of homozygous mutations in *C. elegans* genome. They created not only a strain database that harbor these mutations but also an ortholog-mutation database to predict detrimental mutations in human homologs. The work is one of a kind and tour-de-force. This will be an incredible resource for the *C. elegans* community and the human genetic community. I predict an extremely high citation rate for this manuscript. This certainly deserved to be published in Life Science Alliance. I just have one technical question. From the manuscript, it was not entirely clear whether the majority of the mutants were isolated based on ivermectin resistance. If that was the case, I assume that ivermectin resistance related genes should be vastly overrepresented. I understand that many mutations will come along for the ride but was a little surprised that they would be homozygous. Maybe I did not completely understand how the selection was performed. It might be helpful to spell that out a bit more.

Reviewer #3 (Comments to the Authors (Required)):

This manuscript by Li et al presents TaoChongBao, a valuable resource for the *C. elegans* community. By performing whole-genome sequencing (WGS) on over 12,069 viable EMS-mutagenized strains, the authors have generated a repository of over 541,102 unique missense mutations. This represents a remarkable expansion and supplementation to the previously established Million Mutation Project.

Moreover, the integration of these variants with AlphaMissense pathogenicity predictions and ClinVar clinical annotations creates a powerful "bridge" between in silico predictions and in vivo functional validation. The emphasis on conserved residues between worms and humans makes this a timely tool for interpreting human variants of uncertain significance. The decentralized distribution model proposed is also a pragmatic solution to the logistical challenges of large-scale strain repositories. I recommend this manuscript for publication following minor revisions to address specific technical clarifications and enhance the description of the database.

1. Please incorporate the weblink of TaoChongBao in either the Abstract or the Introduction section.
2. While the authors describe or propose TaoChongBao as a decentralised marketplace enabling laboratories to share resources directly, they did not describe how to do so. If this function is already available, please add a corresponding description in the 'Results' section. If not available yet, please specify this as an upcoming feature.
3. Although EMS mutagenesis is more cost-effective than site-directed mutagenesis by CRISPR/Cas9 technology, please include a brief description of the advantage of CRISPR/Cas9 (allowing for specific codon changes) to give the readers a comprehensive view of mutagenesis technologies.
4. Please include a brief explanation of the distinction between 'predicted pathogenic' and 'clinically reported pathogenic' in the 'Results' or the 'Discussion' section, so that the readers unfamiliar with AlphaMissense and ClinVar can appreciate these annotations more easily.

March 4th, 2026

Life Science Alliance
Manuscript Tracking#: LSA-2026-03631
Authors: Li et al.

Dear Editor,

Thank you for your letter of February 12th, 2026. I now submit our revised manuscript entitled “TaoChongBao: A Large-Scale *C. elegans* Missense Variant Database Bridging Worm and Human Genomes” for publication in Life Science Alliance.

We are glad that reviewers and the journal are interested in the topic and our work. As detailed in response to the reviewers, we have carefully addressed each point by additional experiments, data analysis, and textual clarifications. We appreciate the comments from the reviewers for helping us improve our manuscript.

We thank you for your time and attention and for handling our manuscript.

Sincerely,

Guangshuo

Guangshuo Ou, Ph.D.
Professor
Tsinghua-Peking Center for Life Sciences
School of Life Sciences
Tsinghua University, Beijing 100084, China
E-mail: guangshuoou@tsinghua.edu.cn

RESPONSE TO THE REVIEWERS

Dear Dr. Li,

Thank you for submitting your manuscript entitled "TaoChongBao: A Large-Scale *C. elegans* Missense Variant Database Bridging Worm and Human Genomes" to Life Science Alliance. This manuscript describing a web resource to access and analyze results from the largest ever bank of mutant *C. elegans* lines was reviewed by three expert referees.

As you will see in their reports appended to this letter, all three reviewers voiced clear enthusiasm for this work. Each reviewer made suggestions to improve the text and the associated web platform, and here we draw your attention to the question of ivermectin resistance noted by Reviewers 1 and 2. We invite you to address these and all other suggested improvements in the manner of your choice. Given their unanimous support for publication without major reservations or concerns, we would be happy to publish your paper in Life Science Alliance pending these text changes and final revisions necessary to meet our formatting guidelines.

We appreciate that the Editor and Reviewers are positive to our study.

Reviewer #1 (Comments to the Authors (Required)):

Li, Wang, et al describe a massive collection of viable *C. elegans* mutant strains in which they have used whole genome sequencing to map millions of mutations throughout their genomes. Notably, they have classified all of these mutations based on their presence in coding or non-coding regions, and they have identified over 500,000 mis-sense mutations across nearly all *C. elegans* genes (with an average of 26 mis-sense mutations per affected gene, a remarkable accomplishment). Furthermore, they have established a user-friendly database that integrates the mutation collection with Alpha-fold-based pathogenicity scores and clinical annotations. They also identify a subset of the mutant strains that are ivermectin-resistant and thus likely defective in ciliary or ion channel functions. This is an extremely impressive viable and well described mutant collection and database that are generally available and should provide numerous opportunities for other researchers to explore human disease gene variant function using *C. elegans* as a model system. This manuscript without question warrants publication as written, but the following minor comments and suggestions are offered for the authors' consideration.

We appreciate that the reviewer is positive to our work.

1. Growing viable but somewhat sickly mutant strains can inadvertently lead to the appearance of spontaneous suppressor mutations that improve viability over multiple

generations of growth. It might be worth noting roughly how many generations the mutant strains were grown before being frozen, assuming they were all frozen.

We thank the reviewer's thoughtful comment. In our workflow, mutant strains were not maintained through extended passaging before preservation. After the phenotype was confirmed in the F3 generation, strains were cultured for no more than one to two additional generations before whole-genome sequencing and cryopreservation. Thus, the probability of spontaneous suppressor mutations occurring and being fixed before sequencing and freezing may not be high. We have clarified this point in the Results on page 5, line 5:

"... To reduce the possibility of spontaneous suppressor mutations arising during strain maintenance, we minimized laboratory passaging. Specifically, each mutant line was cultured for no more than one to two additional generations before sequencing and cryopreservation after phenotype confirmation in the F3 generation. ..."

2. The identification of ivermectin-resistant strains is very nice, and the authors might want to include mention of this in their abstract to make these mutant strains more prominent and visible to interested researchers.

We thank the reviewer for this suggestion, and now included it in the Abstract as:

"... Most strains exhibit resistance to the anti-nematode drug ivermectin, while some others display phenotypes like dumpty morphology, uncoordinated movement, multivulva formation and blistered cuticle. ..."

3. I might have missed it, but I did not see anywhere in the manuscript text indicating that the authors have indeed frozen all of these strains, and if the strains are available upon request. If so, this should be mentioned in the paragraph that points out the limitations of the stock center when it comes to handling such a large mutant collection.

We thank the reviewer for this important comment. We have frozen nearly all mutant strains generated in each batch of mutagenesis, totaling more than 12,000 strains, and we are willing to share these resources upon request. However, to maintain such a large collection requires dedicated infrastructure, including two ultra-low-temperature freezers, and a technician responsible for inventory management, routine freezing and recovery. Moreover, due to the scale of the collection, it is not feasible to re-validate the viability of every frozen strain. In fact, we occasionally observed that some strains cannot be successfully recovered after thawing. Our experience highlights the logistical and financial challenges associated with large-scale strain storage and distribution. We agree that this point should be clarified in the manuscript and have now revised the paragraph discussing the limitations of centralized stock on page 11, line 24:

“At present, we have cryopreserved 12,069 *C. elegans* strains, and are willing to share these resources upon request. However, to maintain such a large collection requires dedicated infrastructure, including two ultra-low-temperature freezers, and a technician responsible for inventory management, routine freezing and recovery. Despite these efforts, the scale of the collection makes it impractical to routinely re-test the viability of all frozen strains. Our experience in maintaining this library has highlighted the logistical and financial challenges associated with a centralized repository. Meanwhile, like existing centralized repositories, such as the *C. elegans* Genetics Center or commercial platforms like Addgene, we also face inherent capacity limits and cannot expand indefinitely as the number of unique strains continues to grow. ...”

4. While beyond the scope of this manuscript, it would be interesting for the authors to grow all of their strains at 25°C to see if any might be temperature-sensitive, with enough function at 20°C to make the strains homozygous viable at that lower temperature. Also, it would be very interesting if the authors could identify a few mis-sense mutations that are highly likely to be relevant to human disease and have visible phenotypes, and then see if using CRISPR to create the mutation in an otherwise wild-type background leads to the same visible phenotype. Alternatively, the authors might mention this possible strategy for future research in the Discussion section.

We thank the reviewer’s insightful comment. We do not exclude the possibility that temperature-sensitive strains may exist in our stock; however, since we did not perform any screening specifically for temperature sensitivity, the probability is likely limited. We appreciate this valuable suggestion and will consider incorporating temperature-sensitivity assays in future work to further explore our mutant collection. We have added this idea in the Discussion on page 11, line 1:

“... It is also meaningful to examine whether mutants in the library exhibit additional phenotypes, such as temperature sensitivity, chemotaxis variant, or chemical hypersensitive. This may provide an opportunity to identify new genes and mechanisms contributing to these phenotypes. ...”

For the second comment, we fully agree that introducing selected missense mutations into a wild-type background is a powerful method to establish direct causality between missense mutations and observed phenotypes. In fact, we have previously applied a similar validation in an earlier study (Chai Y., *et al.*, PNAS, 2024). We introduced the human disease-associated mutation KIF1A (R11Q) into *C. elegans* by generating the corresponding *unc-104* (R9Q) mutant using CRISPR-Cas9 genome editing and observed clear phenotypes including coiling, shortened body length, and uncoordinated movement. Although not explicitly stated, the *unc-104* (R9Q) mutation was originally identified from our mutant collection (which was under development at

that time), and its prioritization for genome editing was guided by homologous pathogenicity analysis based on our database, which revealed that *unc-104 (R9Q)* in *C. elegans* and human KIF1A(R11Q) are precisely a pair of homologous pathogenic mutations. This example demonstrates how our platform can not only catalog variants but also suggest disease-relevant candidates for precise functional validation. We have now highlighted this strategy more explicitly in the Discussion on page 11, line 9:

“Beyond serving as a large-scale library of sequenced mutants, our platform provides a systematic framework facilitating disease modeling. By integrating mutagenized *C. elegans* variants with cross-species conservation analysis and human pathogenicity annotations, the library enables the identification of homologous pathogenic mutations affecting conserved residues. A representative example is the *unc-104 (R9Q)* variant (31), which was initially identified within our developing mutant collection and prioritized through homologous pathogenicity analysis linking it to the human disease-associated KIF1A (R11Q) mutation. The introduction of this mutation into clean background worms using CRISPR-Cas9 successfully recapitulated morphology and movement phenotypes. This case illustrates how the integration of large-scale mutagenesis, pathogenicity annotation, and targeted genome editing can transform a mutant resource into a predictive platform for modeling conserved, disease-associated missense variants. We expect that continued application of this platform will facilitate dissection of human pathogenic mutations and aid in identifying therapeutic targets. ...”

Reviewer #2 (Comments to the Authors (Required)):

In this manuscript Li and colleagues used EMS mutagenesis to generate a large number of homozygous mutations in *C. elegans* genome. They created not only a strain database that harbor these mutations but also an ortholog-mutation database to predict detrimental mutations in human homologs. The work is one of a kind and tour-de-force. This will be an incredible resource for the *C. elegans* community and the human genetic community. I predict an extremely high citation rate for this manuscript. This certainly deserved to be published in Life Science Alliance. I just have one technical question. From the manuscript, it was not entirely clear whether the majority of the mutants were isolated based on ivermectin resistance. If that was the case, I assume that ivermectin resistance related genes should be vastly overrepresented. I understand that many mutations will come along for the ride but was a little surprised that they would be homozygous. Maybe I did not completely understand how the selection was performed. It might be helpful to spell that out a bit more.

We appreciate the reviewer is positive to our work. Approximately 70% mutants in our current collection were isolated through ivermectin resistance screening. We agree that this selection strategy leads to enrichment of mutations affecting ivermectin resistance related pathways, and genes involved in drug response are indeed

overrepresented relative to an unbiased mutational spectrum.

Regarding the selection procedure, EMS-treated P0 animals produced F1 progeny carrying heterozygous mutations. F2 progeny were then singled onto individual plates containing ivermectin for selection. Only F2 animals that survived and were able to reproduce under drug treatment were retained. If these animals produced F3 progeny that stably maintained the resistance phenotype, the F3 generation was expanded by one or two additional generations, followed by whole-genome sequencing and cryopreservation. Because EMS mutagenesis can introduce a large number of random mutations in the P0 germline, background mutations segregate independently in the F2 generation. While the ivermectin resistance related mutations are expected to become homozygous under selection, those background mutations may also become homozygous through Mendelian segregation. Consistent with this, the raw variant calling results typically contain thousands of variants per strain (including both homozygous and heterozygous), but after filtering with vcflib tools (see Methods), the number of retained homozygous variants is reduced to only several hundreds to a few thousands. Therefore, although ivermectin resistance related genes are enriched due to the screening, most EMS-induced mutations remain heterozygous, and only a small subset become fixed in isolated lines.

We have added a paragraph in the Discussion on page 10, line 3:

“Although our strain library provides a large mutant resource, it is important to point out its scope and potential biases. Approximately 70% of the strains were isolated through ivermectin resistance screening, while the others were obtained from morphology variants (e.g., dumpy), locomotor defects (e.g., uncoordinated movement), etc., and a smaller set from suppressor screens. As a result, genes involved in ivermectin response and related pathways are expected to be enriched relative to an unbiased mutational spectrum. This enrichment reflects the screening strategies rather than spontaneous selection within the collection. Nevertheless, since EMS induces numerous random mutations, background variants segregate independently during the F2 generation, therefore may also become homozygous. Consequently, aside the ivermectin resistance associated genes, a small subset of background mutations can still become fixed in individual strains. Future expansion of the library using additional selection strategies and non-selective propagation will further broaden the functional diversity of the mutant collection. ...”

Reviewer #3 (Comments to the Authors (Required)):

This manuscript by Li et al presents TaoChongBao, a valuable resource for the C. elegans community. By performing whole-genome sequencing (WGS) on over 12,069 viable EMS-mutagenized strains, the authors have generated a repository of over 541,102 unique missense mutations. This represents a remarkable expansion and supplementation to the previously established Million Mutation Project.

Moreover, the integration of these variants with AlphaMissense pathogenicity predictions and ClinVar clinical annotations creates a powerful "bridge" between in silico predictions and in vivo functional validation. The emphasis on conserved residues between worms and humans makes this a timely tool for interpreting human variants of uncertain significance. The decentralized distribution model proposed is also a pragmatic solution to the logistical challenges of large-scale strain repositories.

I recommend this manuscript for publication following minor revisions to address specific technical clarifications and enhance the description of the database.

We appreciate that the reviewer is supportive for our work.

1. Please incorporate the weblink of TaoChongBao in either the Abstract or the Introduction section.

We thank the reviewer's advice, and have added the weblink of TaoChongBao in the introduction on page 4, line 1:

"... We developed TaoChongBao (<https://oulab.life.tsinghua.edu.cn/homopatho/index.php>) ..."

2. While the authors describe or propose TaoChongBao as a decentralised marketplace enabling laboratories to share resources directly, they did not describe how to do so. If this function is already available, please add a corresponding description in the 'Results' section. If not available yet, please specify this as an upcoming feature.

We thank the reviewer for pointing out this. Establishing a decentralized marketplace is a long-term goal we have proposed, and has been taken into consideration from the very beginning of building the TaoChongBao. Although the platform currently operates in a centralized manner, we are actively advancing its development and gradually add features that facilitate resource sharing. To clarify this, we updated the Discussion on page 12, line 7:

"... TaoChongBao aims to become such a decentralized platform. We are actively advancing its development and gradually add features that facilitate resource sharing, enabling laboratories to share resources directly while reducing redundancy, distributing storage burdens, and promoting a sustainable, community-driven system for genetic research. ..."

3. Although EMS mutagenesis is more cost-effective than site-directed mutagenesis by CRISPR/Cas9 technology, please include a brief description of the advantage of CRISPR/Cas9 (allowing for specific codon changes) to give the readers a comprehensive view of mutagenesis technologies.

We thank the reviewer's advice. We have added a brief description of the advantages of CRISPR/Cas9 in the Introduction on page 3, line 11:

“... Current approaches, such as CRISPR-based genome editing, enable researchers to manipulate the genome of cell lines and animal models with ease and accuracy, thereby generate specific codon changes and individual missense variants. However, these methods typically produce one mutation at a time, and necessitate several months and substantial costs (6,7). ...”

4. Please include a brief explanation of the distinction between 'predicted pathogenic' and 'clinically reported pathogenic' in the 'Results' or the 'Discussion' section, so that the readers unfamiliar with AlphaMissense and ClinVar can appreciate these annotations more easily.

We appreciate the suggestion to make our manuscript more accessible. We have included a short explanation in Results on page 6, line 17:

“... AlphaMissense provides in silico prediction that estimates the probability of a missense variant disrupting protein function based on a deep learning model. It gives probabilistic scores but does not incorporate patient-level clinical evidence. In contrast, ClinVar offers clinically curated interpretations of variant pathogenicity, submitted by laboratories and researchers, and supported by diverse lines of evidence, including segregation data, population frequency, functional assays, and clinical observations. ...”

March 12, 2026

RE: Life Science Alliance Manuscript #LSA-2026-03631R

Dr. Guangshuo Ou
Tsinghua University
School of Life Sciences
Medical Building D217
Beijing 100084
China

Dear Dr. Ou,

Thank you for submitting your Resource entitled "TaoChongBao: A Large-Scale *C. elegans* Missense Variant Database Bridging Worm and Human Genomes". It is a pleasure to let you know that your manuscript is now accepted for publication in Life Science Alliance. Congratulations on this interesting work.

Your manuscript will now progress through copyediting and proofing.

We appreciate your recent correspondence on meeting journal requirements for data deposition. During proofing please amend the Data Availability statement to note that the data underlying this work can be found at Zenodo (<https://zenodo.org/records/18948671> (DOI: 10.5281/zenodo.18948670)). Please include the description you provided of the two datasets accessible at this link: "The first file contains the complete list of mutations for all strains in our mutant collection. The second file contains the homologous pathogenic mutation dataset that we constructed. These datasets represent the main outputs of our study and constitute the underlying data used by our TaoChongBao platform."

In addition, during proofing please also expand the methods section as needed to ensure full details are included on genomic DNA extraction and the DNA sequencing itself.

Your article will publish open access upon publication under a CC-BY license.

DISTRIBUTION OF MATERIALS:

Again, congratulations on a very nice paper. I hope you found the review process to be constructive and are pleased with how the manuscript was handled editorially. We look forward to future exciting submissions from your lab.

Sincerely,
